# Effect of Loading on Wheat Germ Drying in a Batch Fluidized Bed for Industrial Production

**Der-Sheng Chan** [1],*  **and Meng-I Kuo** [2],*

[1]   Department of Information Technology, Lee-Ming Institute of Technology, New Taipei City 243, Taiwan
[2]   Department of Food Science, Fu Jen Catholic University, New Taipei City 24205, Taiwan
*   Correspondence: dschan@ms58.hinet.net (D.-S.C.); 062998@mail.fju.edu.tw (M.-I.K.)

**Abstract:** A high loading production in the manufacturing process of wheat germ (WG) drying is important for reducing the production costs. From a cost perspective, the drying performance become more effective in a batch process when the loading increases. The objective of this investigation was to evaluate the drying performance of WG with different loadings, from 2 to 9 kg, at 120 °C in a fluidized bed dryer. The moisture content, according to the American Association of Cereal Chemists (AACC) method, and the water activity using a thermal hygrometer were measured. The absolute humidity, diffusivity of moisture, and thermal efficiency were analyzed using a mathematical model. An analysis of the dehydration flux demonstrated a linear relationship between dehydration time and WG loading using a fluidized bed dryer. The kinetics of WG drying were observed with a simple exponential model used to match the experimental observation, indicating that the drying rate constant decreases with an increase in WG loading. A linear relationship was obtained between the WG loading and heating time (heating time = −0.212 + 0.577 × WG loading). On this basis, a process optimization was developed for industrial operation, and for predicting the drying performance of WG for industrial-scale production.

**Keywords:** wheat germ; fluidized bed drying; moisture content; dehydration; drying rate constant

## 1. Introduction

Wheat germ (WG) is an important natural source of plenty of nutrients from wheat milling production. An effective drying performance in a batch process is powerful, not only from a purely engineering point of view, but also from a cost perspective. The performance depends on the production method, which enhances the efficiency and productivity of WG drying. It is very important to estimate the marginal productivity of individual batch processes. Manufacturing costs include the material costs, labor costs, and production costs for a batch process. In batch fluidized bed dryers, labor costs account for a large part of the total cost. With high loading, this approach is more effective than small-scale methods, because of the reduction in labor and production costs. The valorization of such an industrial by-product was established based on its use in many food products, as well as on its use as a successful immobilization biocatalyst [1–3]. A comprehensive review of WG separation, stabilization, and food applications has been previously presented [2]. The important health characteristics of WG have also been extensively investigated [4,5]. WG can be used as an addition to bread [6–8] and cake [9]. A major limitation of WG utilization is its rapid spoilage process. Stabilization methods for preventing WG spoilage can be grouped into the following three types: physical, chemical, and biological approaches [2]. The most common physical stabilization methods can be categorized as a number of different treatments, including fermentation [6,7,10,11], microwave [12–14], gamma irradiation [15], infrared radiation [16,17], and fluidization [11,18–21], which limit the WG enzymatic activity. Controlling the water activity (WA) or moisture content (MC) of WG within a desired range

is an important factor when preventing WG spoilage [18]. The MC of WG should be controlled between 5% and 8%, so as to reduce the enzymatic activity and control the water activity, thereby extending the shelf life [19]. Using fluidized bed drying is one of several methods used to enhance the stabilization of agricultural materials [18–24]. Fluidization variables affect the quality of the WG product by influencing the moment, heat, and mass transfer, as well as the evaporation rate of MC, in a fluidized bed dryer (FBD). These common variables include operation conditions (air velocity, set temperature, heating time, WG loading, FBD type, and FBD size) and surrounding environmental conditions (temperature and relative humidity). The effects of the operation conditions on the drying performance in the FBD have previously been investigated at various set temperatures and heating times [18,21,23].

The thermal treatment affects the diffusivity of moisture by increasing the WG temperature and by changing the evaporation rate. The effect of time–temperature combinations on the performance of WG drying has been previously evaluated [21]. The authors predicted the dehydration time and condensation time from the analysis of mass flux during WG drying. The development and optimization of WG drying will not only be invaluable for laboratory studies, but it will also help to determine the operation conditions for industrial-scale production. Fluidized bed drying is an effective drying process, where the WG samples are loaded on a small scale (50–200 g) [22–24] or large scale (2 kg) [20,21]. The bed height increases as the grain loading increases. The effect of the bed height on the drying performance has previously been investigated [25,26]. However, from a commercial point of view, the process of batch FBD with a high WG loading will reduce the drying cost. When the WG drying performance is controlled at a laboratory scale, the most important challenge for the food engineer is the mass production of the drying process. A critical factor of the industrial-scale approach is the identification of the operation conditions, which must be determined from laboratory-scale FBD, in order to replicate the same drying performance. The evolution of the absolute humidity of hot air can be determined from the heat and mass transfer, as well as from the interaction between hot air and cold grain particles [27–32]. Poós and Szabó [33] observed that the margin of error can be reduced by using a modified Nusselt number and a volumetric heat transfer coefficient to study the heat and mass transfer phenomena; useful and practical information regarding the method of the laboratory experiment was obtained in a pilot-plant fluidized bed dryer, in the range of 17 < Re < 1183, with a constant drying rate.

The thermal efficiency of WG drying is a function of the time–temperature history. The relationship between the thermal efficiency and operation conditions has previously been studied during wheat drying in FBD [31,32]. Thin-layer drying curve models describing the variation of moisture with time have been used to evaluate the drying characteristics [34–37]. These models used a drying rate constant to fit the drying curve. The relationship between the drying rate constants and the drying temperature is linear when grain drying with a constant fluidization velocity [35]. The diffusivity of moisture strongly depends on the operation conditions (temperature and time) during grain drying [37–40]. The drying rate was found to decrease over time with WG grains [18] or with onion slices [38] using FBD. The kinetics of WG drying using a diffusion equation have been applied in order to improve the understanding of the WG stabilization process with FBD [18]. However, the drying performance of WG with high loading has not received enough attention regarding its relation to heating time. A mathematical model was developed in our previous study on WG drying [20]. On these grounds, the objective of this work was to develop a mass production strategy to predict the drying performances with different WG loading in an industrial-scale production. Using this information, a process optimization with different WG loadings was developed for industrial operation, and for predicting the drying performance of WG for industrial-scale FBD.

## 2. Materials and Methods

### 2.1. Analytical Methods

The raw WG samples were provided by a local company (Texture Maker Enterprise Co., Ltd., Taiwan) and stored at a temperature of −20 °C. The dimensions of the WG particles were obtained, where the WG had a flat ellipsoidal shape with a diameter of 810 ± 163 um [20]. The raw WG sample was placed in a two-layer polyester package in an environmentally-controlled store room at 25 °C for 12 h, to keep WG temperature constant before the fluidized drying experiment. Raw WG stored at room temperature is not stabilized under an oxygen environment with a high humidity in Asia. The measurements of WA and MC of dried WG product were obtained during drying process in the FBD. The MC of WG was measured using the AACC Method 44-19 [41]. Using the AACC method, 2.0 g ± 1 mg of WG product was dried at 135 °C for 2 h. The WA of the WG was obtained with a thermal hygrometer (Testo 635, Testo Inc., Lenzkirch, Germany). The product temperature was measured using a K-type thermometer (Tecpel 318, Tecpel Co., Ltd., Taipei, Taiwan) at the bottom of the tank supports. The air velocity was obtained by the gas flow transmitter (LABO-FG, GHM Messtechnik GmbH, Erolzheim, Germany). The values of the MC, WA, and heater temperature were obtained from the means of two measurements.

### 2.2. Fluidized Bed Equipment and Process Strategy

In this study, the drying performance with different WG loadings was disposed using the FBD system (Shia Machinery Industrial Co. Ltd., Taichung City, Taiwan). The FBD system was constituted of an air compressor, heater, sample bin, drying chamber, filter bags, and an outlet air motor. A detailed scheme of the FBD system used in this drying measurement was referred to in our previous investigation [20]. In the batch fluidized bed drying preheating, the sample loading, heating, and cooling stages were executed in the drying process. According to our previous study [20], the time allocations of each stage were set and listed on the Table 1. The temperature of the preheated FBD was set at 120 °C for 10 min. Three different weights (2, 4, and 9 kg) of particles of the raw WG sample were put in the sample bin. In the condition of the 9 kg WG loading with an air velocity of 1.2 m/s, the fluidization exists slightly, as a fixed bed behaves. Thus, to prevent acting as a fixed bed, the 2 and 4 kg weights of the raw WG were fluidized with 1.2 m/s, and the 9 kg was fluidized with 1.4 m/s. In this mass production operation, the heating times were used for different WG loadings during the drying in the industrial-scale production. After the WG sample was loaded, the samples were dried at 120 °C with 1.0 min for 2.0 kg, with 2.0 min for 4.0 kg, and 5.0 min for 9.0 kg, respectively. The product temperature was around 45 °C. To reach the product temperature, the process was operated in the condition where the process time was longer than the dehydration time. The surrounding environmental conditions were a relative humidity of 75% and a temperature of 30 °C during the WG drying.

**Table 1.** Time allocation of each stage for wheat germ (WG) drying with 2, 4, and 9 kg.

| Stages (min) | WG Loading (kg) | | |
|:---:|:---:|:---:|:---:|
| | 2 | 4 | 9 |
| Preheating | 10.0 | 10.0 | 10.0 |
| Sample Loading | 1.0 | 1.0 | 1.0 |
| Heating | 1.0 | 2.0 | 5.0 |
| Cooling | 17.0 | 16.0 | 13.0 |

## 3. Model Development

### 3.1. Governing Equation

The governing equations for the microscopic mass and energy balance of the WG, the macroscopic mass and energy balance for emulsion phase, mass balance for bubble phase, and the variable equations were developed (readers can refer to our previous study [20]).

### 3.2. Thermal Efficiency for Wheat Germ Drying

The thermal efficiency, $E_f$, was given so as to evaluate the interaction between air and grain [32], as follows:

$$E_f = \frac{\textit{Latent heat required to evaporate grain moisture}}{\textit{Heat supplied to the drying air}} \tag{1}$$

In this study, the thermal efficiency of the WG drying can be expressed in terms of the operation parameters, as follows:

$$\frac{(X_w - X_{we})}{(X_{wi} - X_{we})} = \exp(-kt) \tag{2}$$

### 3.3. Drying Rate Constant for Wheat Germ Drying

Using the relation between the MC and the drying time, with a simple model based on the first–order kinetic, the drying rate constant can be expressed as follows [31,35]:

$$\frac{(X_w - X_{we})}{(X_{wi} - X_{we})} = \exp(-kt) \tag{3}$$

where $k$ is the drying rate constant. $X_{we}$ is the equilibrium moisture content; a value of 4.7% was set for fitting the experimental data with the simulation response.

## 4. Results and Discussion

### 4.1. Model Verification

The mass transfer coefficient for condensation and dehydration during drying at 120 °C with different WG loadings, from 2 to 9 kg, are given in Table 2. To obtain accurate and reliable results in this study, the criterion of the model verification was calculated using the minimum of the absolute average deviation (AAD) [18] and the determination coefficient ($R^2$), which are shown in the Table 3. The values of AAD were 0.31, 0.19, and 0.31 for the three MC measurements of the WG loadings. The results of the simulated values of the MC were in good agreement with the experimental data, according to the values of the AAD.

**Table 2.** Parameter listed as different WG loading operated; other parameters referred [20].

| Parameter | WG Loading | | |
|---|---|---|---|
| | 2 kg | 4 kg | 9 kg |
| $K_{con}$ (m/s) | $4.5 \times 10^{-3}$ * | $1.2 \times 10^{-3}$ * | $1.5 \times 10^{-3}$ * |
| $K_{de}$ (m/s) | $3.0 \times 10^{-2}$ * | $1.0 \times 10^{-2}$ * | $4.8 \times 10^{-3}$ * |

Superscript *: setting value

**Table 3.** The experimental (Exp.) and simulated (Sim.) moisture content (MC), absolute average deviation (AAD), and determination coefficient ($R^2$) for the WG loading (2, 4, and 9 kg).

| Time (min) | 2 kg | | 4 kg | | 9 kg | |
|:---:|:---:|:---:|:---:|:---:|:---:|:---:|
| | Exp. | Sim. | Exp. | Sim. | Exp. | Sim. |
| 0 | 14.45 | 15.00 | 14.8 | 15.00 | 14.57 | 15.00 |
| 2 | 6.56 | 6.72 | 9.69 | 9.51 | 11.02 | 11.61 |
| 4 | 4.99 | 4.26 | 6.55 | 6.06 | 8.73 | 8.50 |
| 6 | 4.52 | 4.18 | 5.62 | 5.65 | 6.88 | 6.58 |
| 9 | 4.60 | 4.22 | 5.59 | 5.69 | 5.69 | 5.38 |
| 12 | 4.58 | 4.46 | 5.67 | 5.81 | 5.33 | 5.42 |
| 15 | 4.74 | 4.70 | 5.66 | 5.92 | 5.37 | 5.58 |
| 18 | 4.99 | 4.85 | 5.87 | 6.02 | 5.42 | 5.75 |
| ADD | 0.31 | | 0.19 | | 0.31 | |
| $R^2$ | 0.996 | | 0.994 | | 0.992 | |

*4.2. Effect of WG Loading on Drying Performance*

4.2.1. Measurement and Prediction of Inlet Air Temperature for FBD

From our previous study, the air temperature in the FBD was strongly dominated by the heater temperature [20,21] during the WG drying. The thermal treatment of the WG was investigated in the FBD, using a constant heating temperature ranging from 90 to 150 °C during WG drying [18]. In this study, the heating temperature is a function of the drying time. The profiles of the heater temperature must first be obtained from the measurement and prediction. The experimental data and the predicted values of the heater temperature during drying with different WG loadings during WG drying are shown in Figure 1. The solid line is a simulated curve of the heater temperature during the WG drying, according to the heater function in our previous study [20]. The predicted results were consistent with the measurement data during drying at 120 °C with 1.0 min for 2.0 kg, 2.0 min for 4.0 kg, and 5.0 min for 9.0 kg loading, respectively. The inlet condition of the FBD can be used for the simulation of inlet air temperature using the profiles of the heater temperature.

4.2.2. Measurement and Simulation of Moisture Content of Wheat Germ

The diameter of the porous mesh holder at the bottom of the sample bin was small so as to enhance the height of the fluidization zone [21]. In this study, the wide design of the sample bin was the same as the one in our previous study [20]. The thermal input of the WG drying can be reduced with a short heating time approach (e.g., 4 min at 80 °C). Thus, with the shorter heating time and higher drying temperature, the drying performance could be enhanced. In this section, the heating temperature was set at 120 °C, and the heating time was shorter than that in our other study [20]. Figure 2A illustrates the measured data and the simulated curves of the MC of the WG during drying with different WG loading from 2 to 9 kg. From the figure there were three different periods—the abrupt decreased period (0–4 min), the equilibrium period (4–12 min), and the slight increased period (12–18 min), with a WG loading of 2–4 kg. The data indicated that the MC first decreased and then increased with the drying time. Figure 2A indicates the behavior of the dehydration and adsorption of th WG product due to moisture evaporation and condensation. When the heating time was shorter than the cooling time, the behavior of the moisture condensation was similar to that which had been reported in our previous study [20]. The experimental results revealed that the MC of the WG with 2 kg of WG loading during drying at 120 °C drastically decreased to a minimal value of 4.52% at the heating stage. With the constant heating temperature ranging from 90 to 150 °C, an abrupt decrease of the MC with time had been reported as a similar trend [18]. From Figure 2A, an operation with a higher heating time was necessary in order to obtain the same MC when the WG loading increased. The MC of the WG became an equilibrium value upon prolonged drying when the WG loading was 9 kg for a heating

time of 5.0 min. Moreover, the results in Figure 2B revealed that the MC of WG was a function of the WG loading during drying. The MC of the WG first decreased and then increased with drying time for 2.0 and 4.0 kg of loading. Figure 2A shows that a minimum value of the MC exists. With the WG loading of 2 kg, the MC increased as the drying time increased after the equilibrium period. The relationship (MC = 0.03 × (drying time − 6) + 4.52; $R^2$ = 0.79) between the MC and the drying time after the equilibrium period was obtained. The drying time was expressed with the minute unit in the above equation. However, when 9 kg of WG was loaded, the MC of the WG decreased in the whole drying time (0–18 min). The rate of weight loss of the WG dried at 120 °C with various WG loadings is shown in Figure 3. The rate of weight loss with a 9 kg loading was higher than that with 2 and 4 kg of loading in the beginning (2 min). In Figure 3, the area under the curve of the mass loss over time for 9 kg of loading was about 4.5 times greater than the area for 2 kg of loading. The effect of solid loading, 50 to 150 g, on the drying rate was negligible for the sprouted wheat with a thermal temperature at 50 °C and an air flow rate of 5.93 m/s [22]. When the WG was dried for 2–9 min, the rate of weight loss with 9 kg of loading was higher than that with 2 and 4 kg of loading. The result indicates that the rate of weight loss depends on the WG loading.

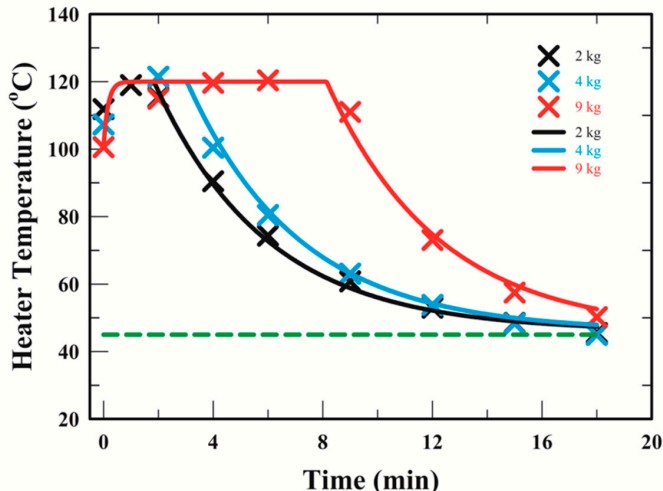

**Figure 1.** Simulated profiles of the heater temperature during wheat germ (WG) drying. The symbols represent the experimental data; the solid lines represent the simulated values; the dashed line represents the product temperature of 45 °C.

**(A)**

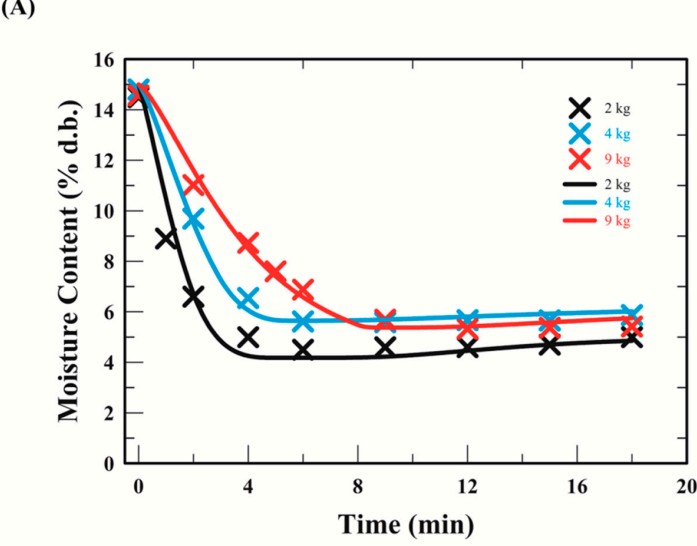

**Figure 2.** *Cont.*

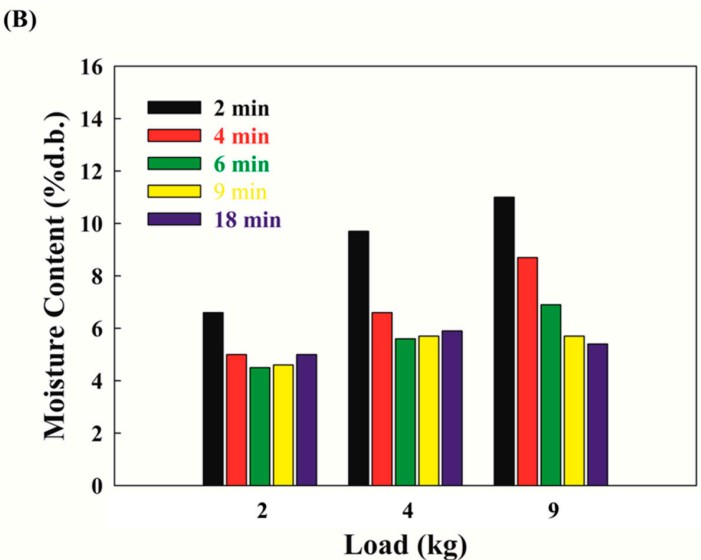

**Figure 2.** Profiles of the moisture content (**A**) and the changes in the moisture content (**B**) of wheat germ with different WG loadings. The symbols represent the experimental data; the lines represent the simulated values.

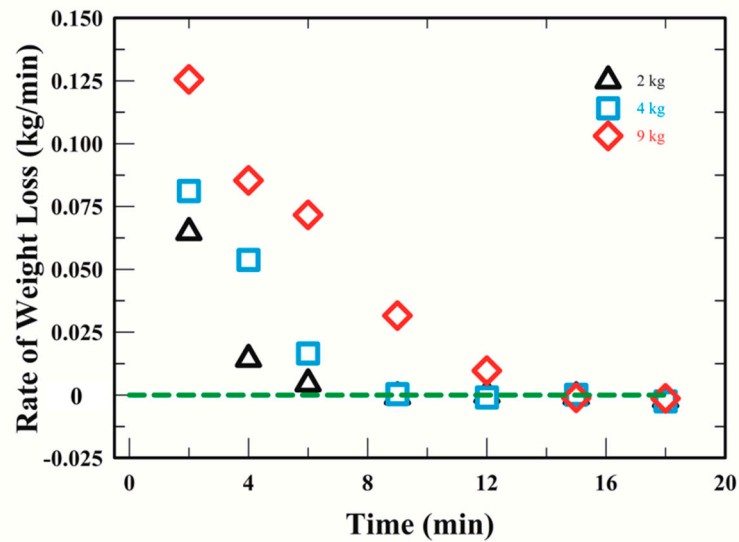

**Figure 3.** Effect of WG loading on the weight loss in a drying procedure of wheat germ by different loadings from 2 to 9 kg at 120 °C. with backward differencing from the data of the moisture content.

4.2.3. Simulated Response of Moisture Distribution in Wheat Germ

The simulation results of the MC of the WG center and the WG surface during drying with different WG loadings, from 2 to 9 kg, in FBD are depicted in Figure 4A. The drying kinetics showed a slower drying rate of the WG center compared with the WG surface at the initial stage of drying. As the drying time goes on, the MC of the WG surface was higher than that of the WG center. This is due to the fact that as the drying time increases, condensation phenomena of the water vapor on the WG surface occurred [20]. When the heating time was shorter than the cooling time, and the heater temperature approached the product temperature, the adsorption of the WG product was observed as a result of moisture condensation. Figure 4B shows the variation of the MC difference between the WG surface and WG center during drying with three different WG loading in the FBD. The value of the MC difference first decreased rapidly, and then increased to zero with the drying time. The similar trend can be obtained with the three WG loadings during drying. The condensation time on the WG

surface in the cooling stage depended on the time–temperature combinations [21]. The result indicated that the adsorption of the WG product is attributed to the moisture condensation. In the meantime, it was observed that the MC difference changes were inconspicuous in the cooling stage.

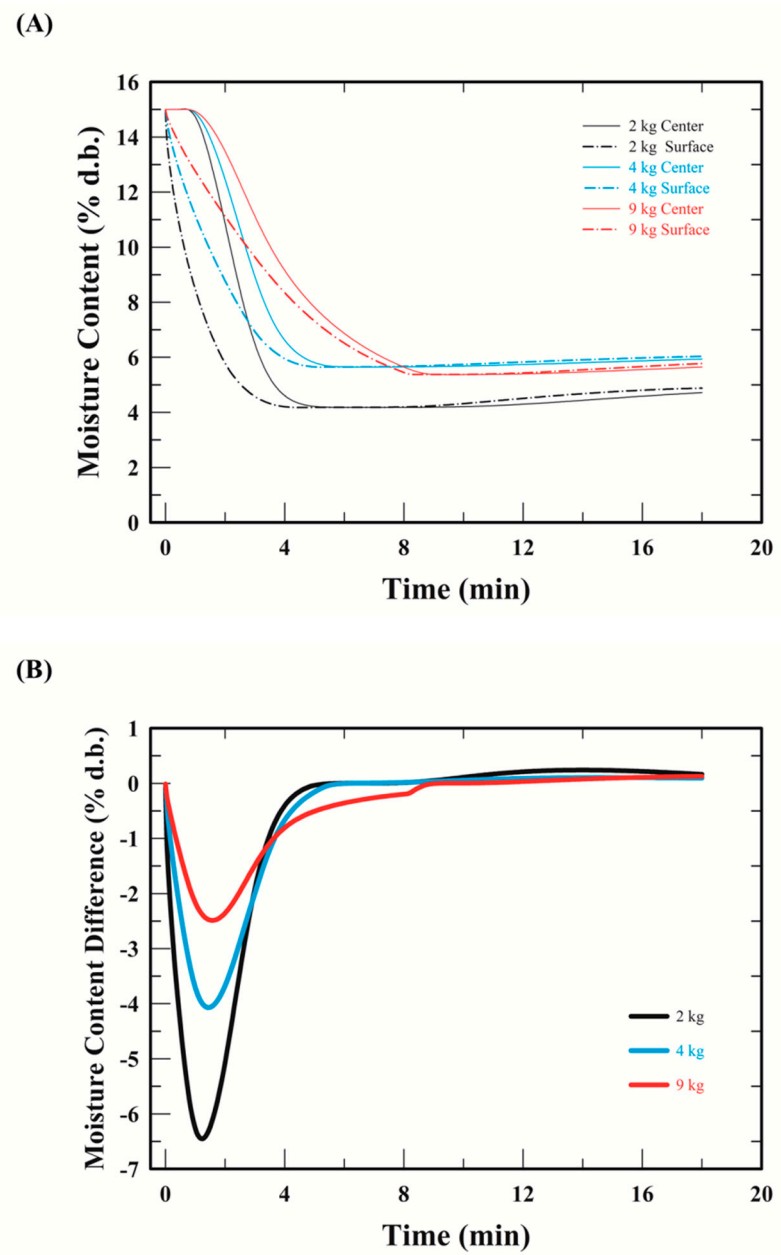

**Figure 4.** Simulated profiles of the moisture content (**A**) and the moisture content difference (**B**) of wheat germ during drying with different WG loadings. Solid lines represent the wheat germ center; dash-dot lines represent the wheat germ surface.

### 4.2.4. Simulated Response of Absolute Humidity

If the absolute humidity of the hot air increases largely, the air becomes saturated during the cooling stage. Figure 5 shows the simulated absolute humidity in the FBD chamber for various WG loadings with inlet air at 30 °C and relative humidity of 75% (absolute humidity is about 0.02 kg/kg). At the initial stage of drying, the absolute humidity of the hot air increases when the drying time increases. As time goes on, the humidity decreases during the WG drying. It is apparent that there is a maximum humidity during the WG drying. This result was also obtained because the rate of

the evaporation of moisture from the corn grains decreases [30]. There was a high increase in the absolute humidity of the hot air with an increased WG loading. In Figure 5, the maximum humidity was 0.045, 0.060, and 0.102 kg/kg with 2, 4, and 9 kg WG loadings during the drying, respectively. The time for the humidity to reach the maximum were about 3.1, 3.9, and 7.4 min with 2, 4, and 9 kg WG loading, respectively. With the same sample bin in an FBD, the bed height increased as the WG loading increased. Hemis et al. [27] observed that the higher absolute humidity of the air was obtained as the bed depth increased from 1 to 10 cm for wheat drying. A similar effect exists on the humidity response with both the grain loading and the bed height.

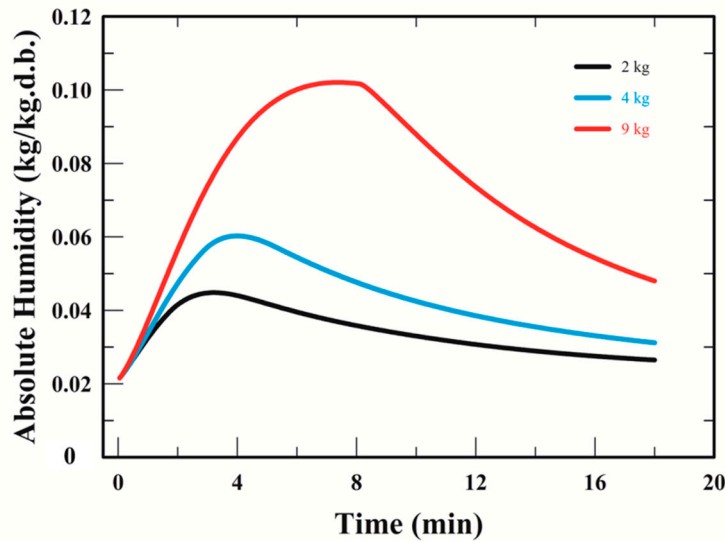

**Figure 5.** Simulated results of the change in humidity of the hot air in the fluidized bed dryer (FBD) during drying with different loadings, from 2 to 9 kg, at 120 °C.

### 4.2.5. Simulated Response of Dehydration Flux and Condensation Flux

In the study of [20], there is the term $K_{con}(C_{me} - C_{sat}) * f_{con}$, where $C_{sat}$ is the saturation concentration, which was determined so as to evaluate the condensation effect in the cooling stage. When the air temperature decreased, the moisture concentration ($C_{me}$) in the emulsion phase might be higher than the saturation concentration on the WG surface in an FBD with the design of seven filter bags. The dehydration time provides significant information to evaluate the effective dehydration in the heating procedure [21]. The dehydration time could be obtained when the dehydration flux was zero. In order to investigate the effect of the WG loading on the dehydration and condensation, the analysis of mass flux is shown in Figure 6A. An additional figure was depicted in Figure 6B, illustrating the mass flux in the cooling stage. The values of the dehydration time were 4.8, 5.3, and 8.6 min during drying at 120 °C, with heating 1.0 min for 2.0 kg, 2.0 min for 4.0 kg, and 5.0 min for 9.0 kg loading, respectively. The effect of prolonged heating could be obtained from the difference between the dehydration time and the heating time. The prolonged time with WG loadings of 2, 4, and 9 kg were obtained 3.8 min (4.8–1.0 min), 3.3 min (5.5–2.0 min), and 3.6 min (8.6–5.0 min), respectively. The mean of the prolonged time was approximately constant (average 3.5 min). The information from the value of the prolonged time is an important key to diagnosing the drying performance of the FBD.

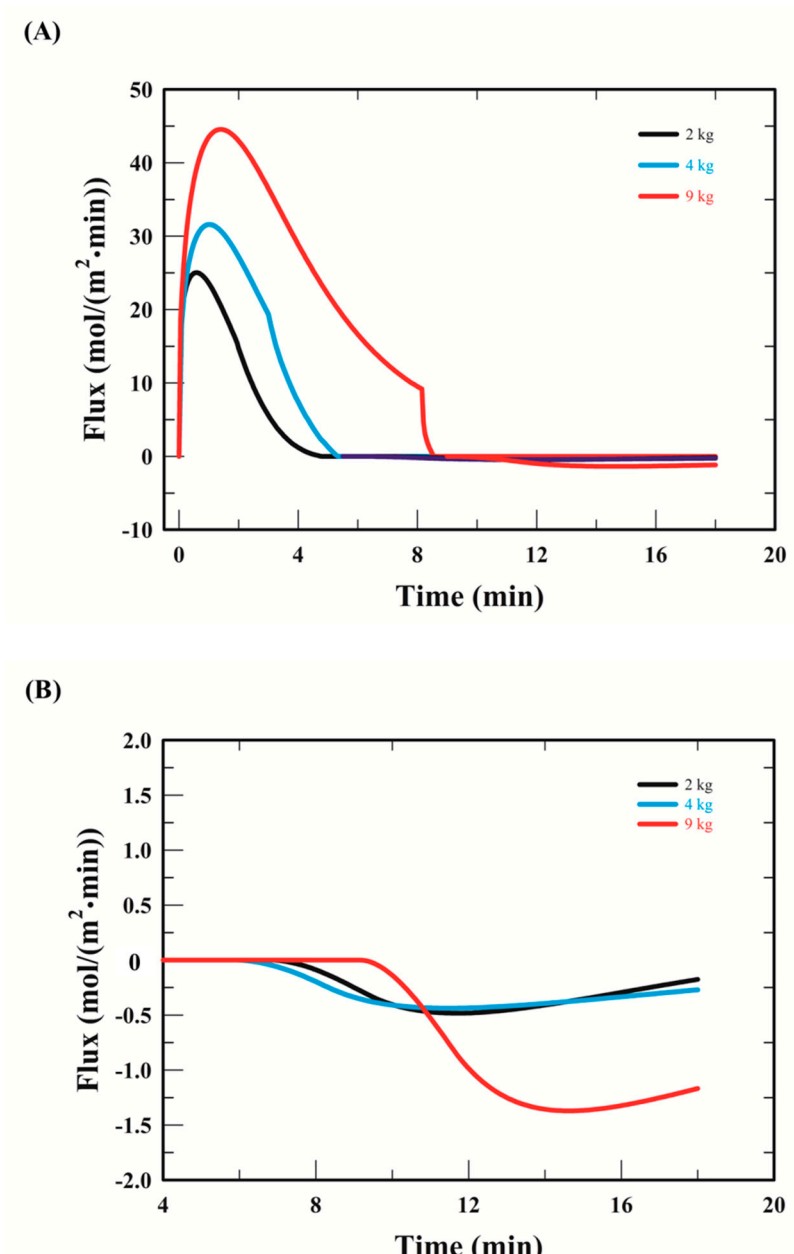

**Figure 6.** Simulated profiles of the mass flux on the WG surface in the heating stage (**A**) and cooling stage (**B**) during drying in the FBD with different WG loadings, from 2 to 9 kg, at 120 °C.

### 4.2.6. Average Diffusivity of Moisture in Wheat Germ

During the wheat drying in an FBD, the values of moisture diffusivity, varying from 7.3 to $30.4 \times 10^{-10}$ m$^2$/s, increased as the drying temperature increased within temperatures ranging from 40 to 66 °C [22]. The effective diffusivity with various air temperatures, ranging from 90 to 150 °C, was determined at an interval of $3.22 \times 10^{-11}$ to $2.38 \times 10^{-10}$ m$^2$/s, by an analytical solution of a thin layer diffusion model during WG drying [18]. The diffusivity of the moisture strongly depends on the time–temperature history of the WG drying. The simulation results of the moisture diffusivity during the WG drying for different WG loadings are shown in Figure 7. It was found that the diffusivity first increased and then decreased in response to increasing the drying time. From the beginning of the drying to the dehydration time, the average diffusivity of the moisture inside WG was $1.34 \times 10^{-8}$, $1.56 \times 10^{-8}$, and $2.10 \times 10^{-8}$ m$^2$/s for the 2, 4, and 9 kg WG loadings, respectively. The values of the average diffusivity were higher when the WG loading increased. It was observed that the diffusivity of

moisture in WG was higher when the WG was dried with a higher time–temperature combination [21]. The value of the diffusivity obtained in this study is higher than the value of diffusivity from the study of Gile et al. [18].

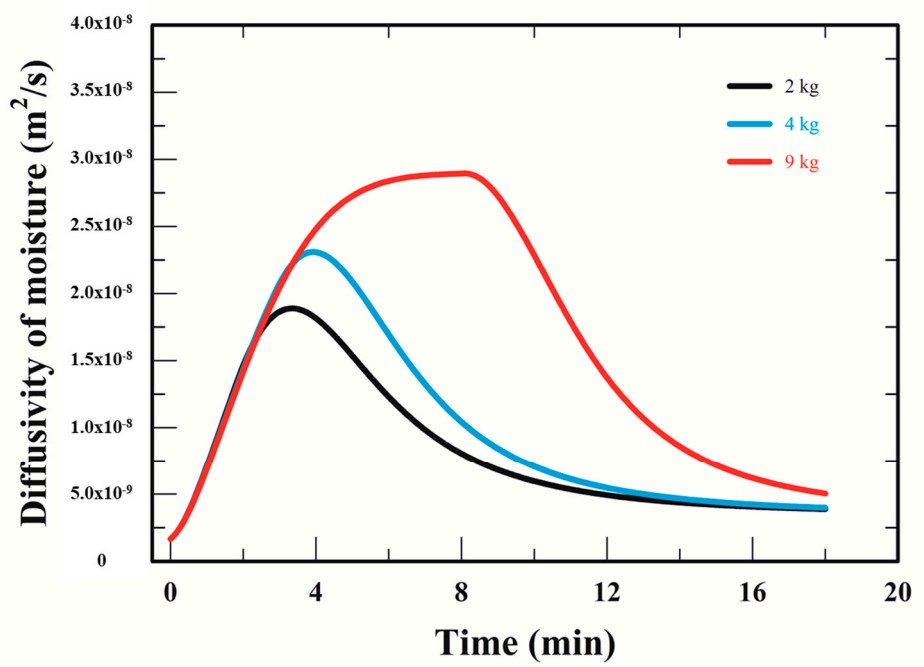

**Figure 7.** Simulated results of the change in the diffusivity of moisture during drying with different loadings, from 2 to 9 kg, at 120 °C.

### 4.2.7. Simulated Response of Water Activity

The water activity is the ratio of the vapor pressure of the food moisture divided by the saturation vapor pressure of the pure water at the same temperature (25 °C). The simple linear relation between the WA and MC (WA = 0.045 × MC + 0.07) was obtained during the WG drying [21]. Figure 8A illustrates the measured data and the fitting curves of the WA of the dried WG during drying with different WG loadings from 2 to 9 kg. The predicted WA of the dried WG was in good agreement with the experimental measurement. It was observed that the WA of the WG product was extremely close to 0.3 with the three WG loadings when the air temperature reached the product temperature (45 °C). Furthermore, the WA of the WG product increased with the WG loading when the heating time was kept constant (Figure 8B). For example, the water activity of the WG loaded with 9 kg was higher than that of the WG loaded with 2 kg when the WG was dried for 2 and 4 min, respectively.

### 4.2.8. Effect of WG Loading on Dehydration time, Heating time and WA

From a cost perspective, it had an effective drying performance in a batch process when the WG loading increased. It is important to obtain the heating time for the WG drying when the WG loading increases, in practice. The effects of the WG loading on the heating time have not received enough attention. Thus, in this study, the relation between the loading and heating time was investigated. The relation between the WG loading, heating time, and dehydration time, and water activity are shown in Figure 9. The dehydration time increased as the WG loading increased from 2 to 9 kg. The following fitting curve was obtained: dehydration time = 3.4 + 0.565 × WG loading. It was observed that a linear relation exists between the WG loading and heating time (heating time = −0.212 + 0.577 × WG loading). The lines of the dehydration time and heating time were almost parallel lines. In the meantime, the water activity of the WG could be maintained within a target range of 0.3 ± 0.1, which

was able to attain the condition of the WG stored at a water activity of 0.4–0.5, so as to avoid spoilage from the enzymatic activity and lipid quality losses [16].

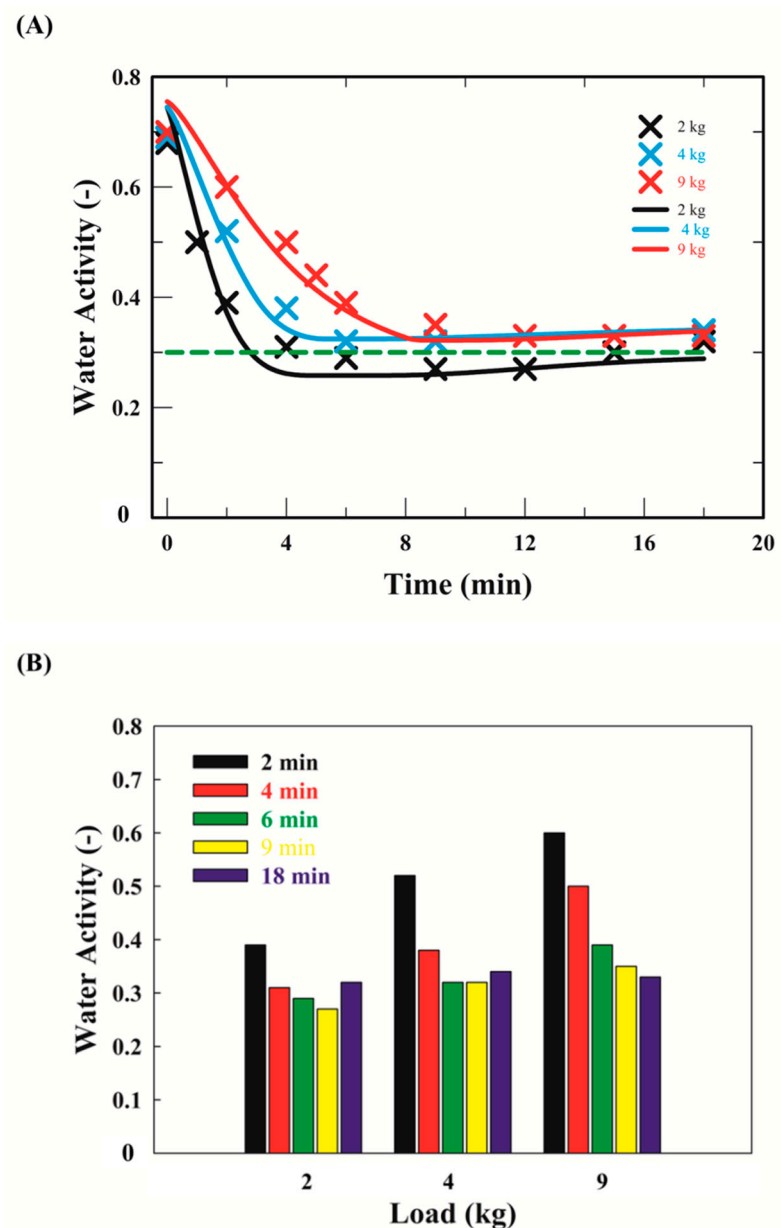

**Figure 8.** Profiles of the water activity (**A**) and the changes in the water activity (**B**) of wheat germ during drying with different WG loadings. The symbols represent the experimental data; the lines represent the simulated values.

### 4.2.9. Simulated Response of Heating Efficiency

The effect of the bed height on the drying performance of the grain had been studied [25,26,31,32]. Giner and Calvelo [31] pointed out that the thermal efficiency increases and the residence time of the grains increase with a rise in bed height. It is apparent that bed height increased when the WG loading increased, using the same sample bin in an FBD. From the basis of these wheat drying studies, the effect of the WG loading on the thermal efficiency of the WG drying in an FBD was evaluated in this investigation. In Equation (2), the thermal efficiency is a function of the heating time. The denominator term is the energy input used to evaporate the water into the air from a heater. The simulation results of the thermal efficiency for the three WG loadings, from 2 to 9 kg, with the drying time are shown in

Figure 10. These results show the thermal efficiency as a function of the drying time and WG loading. According to the calculation using Equation (2), the thermal efficiency decreases as the drying time increases. The efficiency increased in response to the increased WG loading. This relation between efficiency and WG loading is in agreement with the results in the literature [31,32]. It is reasonable that the operation cost decreased when the WG loading increased in a batch FBD.

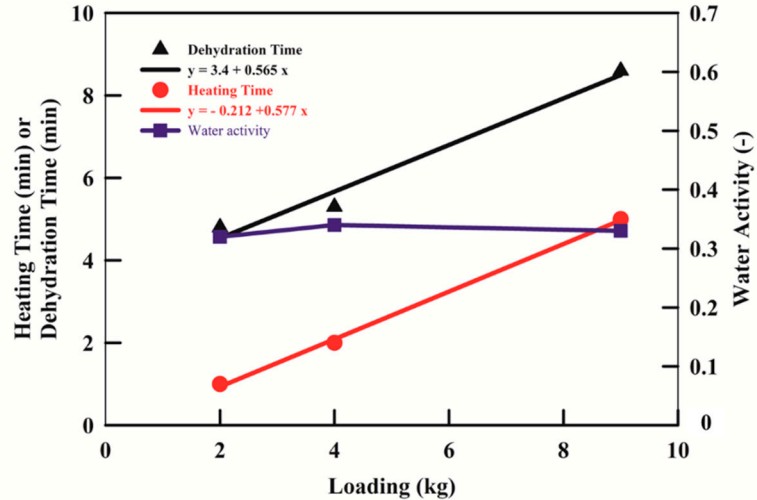

**Figure 9.** Effect of WG loading on the dehydration time, heating time, and water activity with different loading, from 2 to 9 kg, at 120 °C. Dehydration time (▲), heating time (●), and WA (■); the lines represent the fitting value.

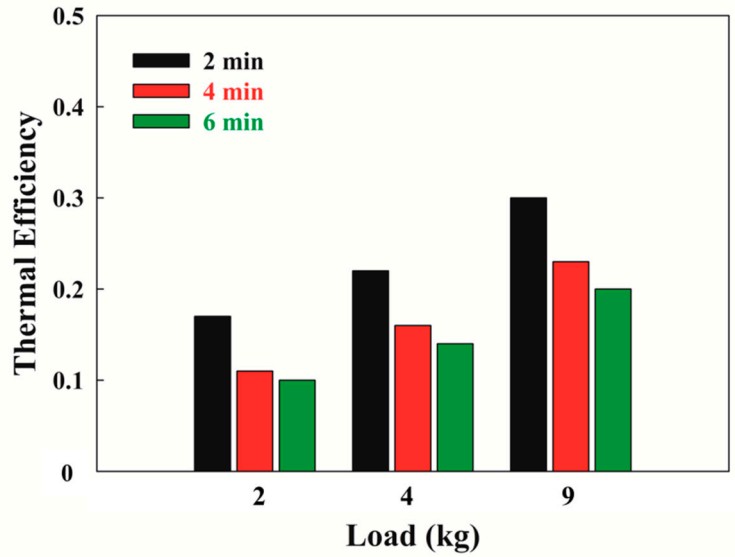

**Figure 10.** Thermal efficiencies as a function of the WG loading during drying with different loadings, from 2 to 9 kg, at 120 °C.

### 4.2.10. Simulated Drying Kinetics with Simple Exponential Model

Knowledge of the drying kinetics was used to determine whether the drying performance was operated in an ideal batch FBD. A comparison between the predicted and experimental values of MC with different WG loadings, from 2 to 9 kg, is shown in Figure 11. The value of the determination coefficient ($R^2 = 0.997$) of this curve indicated that the predicted model is a good fit for the actual drying data. Bebartta et al. [38] indicated that the drying rate is constant with different drying conditions (air temperature, air velocity, and onion slice thickness) for fluidized bed drying. The results indicated that the values of the drying rate constants decreased with the increase in thickness of onion slice

at a constant drying air temperature and velocity. Giner et al. [31] studied the effect of the different bed heights and air temperatures on the wheat drying performance with a fluidized bed. The results indicated that the drying time for attaining a given grain moisture increased when the bed height increased. Figure 12 shows that the drying rate constant decreased in response to the increased WG loading. The results indicated that the drying rate constant decreased as the WG loading increased from 2 to 9 kg. With a high WG loading in an FBD, it is convincing that a partial degree of fluidization existed.

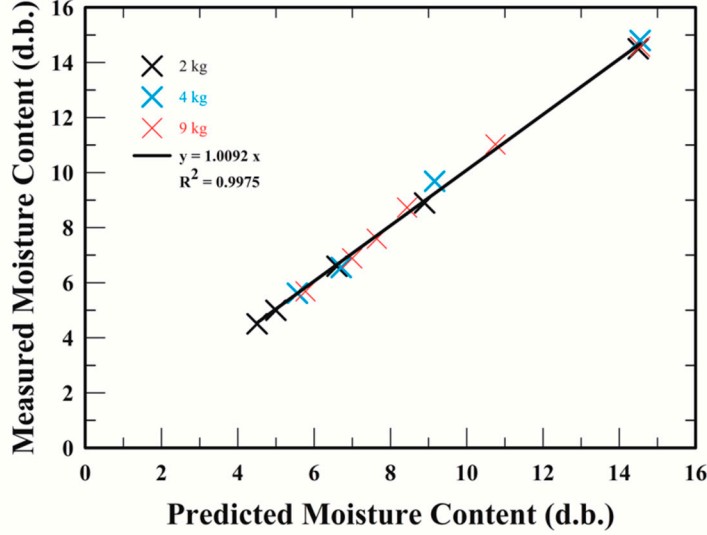

**Figure 11.** Experimental values of the moisture content versus the predicted values using a simple exponential model for WG drying with different loadings, from 2 to 9 kg, at 120 °C; the lines represent the fitting value.

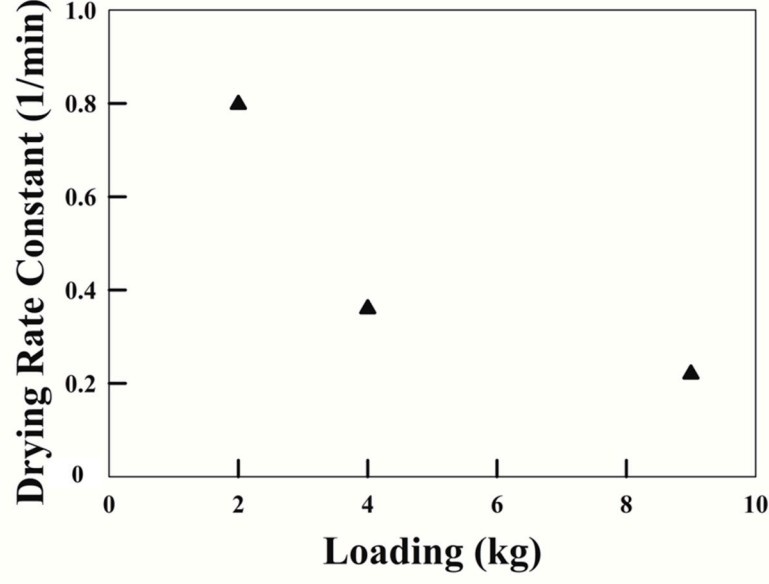

**Figure 12.** Drying rate constant versus WG loading during drying with different loadings, from 2 to 9 kg, at 120 °C.

## 5. Conclusions

This study considered a cost perspective with an effective drying performance in a batch process. The drying performance of wheat germ with different loadings, from 2 to 9 kg, at 120 °C in an FBD has been studied by the prediction of model and experimental results. The moisture content, water

activity, dehydration flux, thermal efficiency, and drying rate were analyzed in the function of time with different WG loadings. From the analysis of the dehydration flux, it was observed that a linear relation existed between the WG loading and the heating time. The thermal efficiency increased as the WG loading increased from 2 to 9 kg. From the drying kinetic analysis, the result indicated that the drying rate constant decreased sharply as the WG loading increased. This is sensible, as a partial degree of fluidization existed when the WG loading increased. The information and approach from this study are useful in scaling up the drying process of industrial dryers.

**Author Contributions:** D.-S.C. performed the experiments and developed the research ideas, process model, and computational simulations. M.-I.K. was responsible for the conception, research ideas, and research grant.

**Funding:** This research was funded by Texture Maker Enterprise Co., Ltd., Taiwan, grant number 600312.

**Conflicts of Interest:** The authors declare no conflict of interest.

## Nomenclature

**Symbol Meaning (units)**

| | |
|---|---|
| $C_{me}$ | moisture concentration in the emulsion phase (mol/m$^3$) |
| $C_{sat}$ | saturation concentration (mol/m$^3$) |
| $C_{p,g}$ | specific heat of air (J/(kg K)) |
| $E_f$ | thermal efficiency ($-$) |
| $f_{con}$ | step numerical parameters for condensation |
| $k$ | drying rate constant (1/s) |
| $K_{con}$ | mass transfer coefficient for condensation (m/s) |
| $K_{de}$ | mass transfer coefficient for dehydration (m/s) |
| Re | Reynolds number ($-$) |
| $S$ | bed section (m$^2$) |
| $T_a$ | air temperature (K) |
| $T_{ev}$ | environmental temperature (K) |
| $t$ | time (s) |
| $V_0$ | air velocity (m/s) |
| $W_l$ | loading of wheat germ (kg) |
| $X_w$ | moisture content of wheat germ (%; d.b.) |
| $X_{we}$ | equilibrium moisture content of wheat germ (%; d.b.) |
| $X_{wi}$ | initial moisture content of wheat germ (%; d.b.) |

**Greek symbols**

| | |
|---|---|
| $\rho_g$ | air density (kg/m$^3$) |
| $\lambda$ | latent heat of vaporization (J/kg) |

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
