# Peer review of "Effect of Loading on Wheat Germ Drying in a Batch Fluidized Bed for Industrial Production"

_processes, doi:10.3390/pr7120864_

Round 1

Reviewer 1 Report

The review of the submission is attached.

Author Response

The attached files are the responses to the reviewers. We appreciate your time.

Reviewer 2 Report

Manuscript entitled ‘Effect of Loading on Wheat Germ Drying in Batch Fluidized Bed for Industrial Production’ presented an evaluation of the drying performance of wheat germ with different loading 2-4-9 kg at drying temperature 120 °C in fluidized bed dryer by received experimental results and the model’s prediction.

Suggestions for Authors:

Fluidized bed drying, how many repetitions were made of drying for wheat germ? It should be mentioned in the manuscript.

How many repetitions were made for analysed properties of samples? Was statistical analysis performed? The manuscript should contain the number of repetitions made. Nothing mentioned!

What was the wheat germ storage temperature (-20 oC) dictated? Explain why the material was not stored at room temperature?

Author Response

(The authors gave the same response as above.)

Reviewer 3 Report

The manuscript "Effect of Loading on Wheat Germ Drying in Batch luidized Bed for Industrial Production" describes the process optimization of wheat drying. The manuscript is well designed and disccussed; however there are several aspects which must be adressed prior to its acceptance.

M&M section does not describe clearly all the determinations that authors report. Sub sections of M&M must be clarified or at least properly referenced.

Since authors want to validate a mathematical model, authors must be more rigorous and perform statistical analysis in order to obtain more accurate and reliable results.

Once authors have included the statistical analysis in both M&M and R&D section manuscript will be ready for its acceptance.

The manuscripts needs major revisions.

Author Response

(The authors gave the same response as above.)

Reviewer 4 Report

Please find attached comments for the authors

Author Response

(The authors gave the same response as above.)

Reviewer 5 Report

Authors in the present study have studied the drying performance of wheat germ by different loadings (2 - 9 kg/120 °C) in a fluidized bed dryer. It is an interesting study with good perspectives as this process may interest the industry. The general presentation of the study is very good only some minor aspects need to be addressed prior publication.

Authors could also mention in the introduction part other small scale drying methods of wheat germ and compare advantages and disadvantages with this particular method by a cost point of view and a scientific point of view. Does this method influence the quality of wheat germ? Could freeze-drying be used as an effective method despite the high costs? More data regarding wheat germ food applications could be mentioned. Valorization of such industrial by-product has been utilized in many food products and can also be used as a successful immobilization biocatalyst [1-3]. Also important health characteristics of wheat germ could be mentioned [4,5] (Lines 28-29). Line 160: Please add the Figure caption and revise all other figures accordingly. 4: Please add the specific different WG loadings Line 298-300: Please revise grammar.

References

Mantzourani, I.; Terpou, A.; Alexopoulos, A.; Bezirtzoglou, E.; Bekatorou, A.; Plessas, S. Production of a potentially synbiotic fermented Cornelian cherry (Cornus mas L.) beverage using Lactobacillus paracasei K5 immobilized on wheat bran. Biocatalysis and Agricultural Biotechnology 2019, 17, 347-351, doi:10.1016/j.bcab.2018.12.021. Boukid, F.; Folloni, S.; Ranieri, R.; Vittadini, E. A compendium of wheat germ: Separation, stabilization and food applications. Trends in Food Science & Technology 2018, 78, 120-133, doi:https://doi.org/10.1016/j.tifs.2018.06.001. Mantzourani, I.; Terpou, A.; Alexopoulos, A.; Bezirtzoglou, E.; Plessas, S. Assessment of Ready-to-Use Freeze-dried Immobilized Biocatalysts as Innovative Starter Cultures in Sourdough Bread Making. Foods 2019, 8, 40. Călinoiu, L.F.; Cătoi, A.-F.; Vodnar, D.C. Solid-State Yeast Fermented Wheat and Oat Bran as A Route for Delivery of Antioxidants. Antioxidants 2019, 8, 372. Yu, L.; Zhou, K.; W Parry, J. Inhibitory effects of wheat bran extracts on human LDL oxidation and free radicals. LWT - Food Science and Technology 2005, 38, 463-470, doi:https://doi.org/10.1016/j.lwt.2004.07.005.

Author Response

(The authors gave the same response as above.)

Round 2

Reviewer 1 Report

The introduction contains the following text in the revised version of the manuscript:

The critical point of the industrial-scale approach is the identification of the operation conditions which must be useful between the laboratory and the industrial-scale FBD, in order to replicate the same drying performances. The evolution of the absolute humidity of hot air determined by heat and mass transfer and interaction between hot air and cold grain practices [27-32]. The are many useful practical information about the method of the laboratory experiments with fluidized bed dryers reported in [*]

[*]: „Volumetric Heat Transfer Coefficient in Fluidized‐Bed Dryers” (https://doi.org/10.1002/ceat.201700038).

It would be nice if the Authors complete their manuscript referring to this publication, because this topic is very close to their studies.

Author Response

Dear academic editor and reviewer:

We very appreciate your comments and suggestions. The following is our responses to your comments.

Reviewer 3 Report

Manuscript has been improved according to reviewers' recommendation and must be accepted in its present form.

Author Response

(The authors gave the same response as above.)
